# Improving pediatric care in Uganda with a digital platform and quality improvement initiative: A retrospective review of Smart Triage + QI

Rebecca Goertzen[1]*, Yashodani Pillay[1,2], James Karugaba[3],
Ivan Aine Aye Ishebukara[3,4], Ahmad Asdo[1,2], Dustin Dunsmuir[1,2], Justine Behan[1],
Charly Huxford[1], Stefanie K. Novakowski[1,2], Fredson Tusingwire[4], Ronald Kasyaba[5],
Gloria Kakuru[5], John Khisa[5], Stephen Businge[3], Mike Kyewalyanga[3],
Niranjan Kissoon[2,6], J. Mark Ansermino[1,2]

**1** Institute for Global Health at BC Children's and Women's Hospital+Health Centre, Vancouver, BC, Canada, **2** Department of Anesthesiology, Pharmacology & Therapeutics, University of British Columbia, Vancouver, BC, Canada, **3** Holy Innocents Children's Hospital, Mbarara, Uganda, **4** Walimu, Kampala, Uganda, **5** Uganda Catholic Medical Bureau, Kampala, Uganda, **6** Department of Pediatrics, University of British Columbia, Vancouver, BC, Canada

* rebecca.goertzen@cw.bc.ca

## Abstract

### Objective

This is a retrospective review of the feasibility study and implementation of the Smart Triage and Quality Improvement (QI) initiative at Holy Innocents Children's Hospital (HICH), a dedicated pediatric hospital in Mbarara, Uganda, over a 5-year period. The aim of this QI initiative was to improve triaging rates and the time-to-antimicrobials in HICH's outpatient department (OPD).

### Methods

Smart Triage is a risk prediction algorithm and digital platform that enables healthcare workers to triage patients and track treatments effectively. Following the feasibility study, the QI program was implemented in September 2021 using three Plan-Do-Study-Act cycles: 1) Standardize Training, 2) Adjust Workflows, and 3) QI Team Communication. Data sources were triage and hospital reports. Monthly run charts of OPD attendance, acuity of illness, triaging rates, median-time-to-antimicrobials, and mortality rates of admitted patients were created. The trajectories of the variables were assessed using linear regression with time as the explanatory variable.

### Results

121,521 children attended HICH OPD from November 2018 to October 2023. The OPD triaging rate increased to 91% by October 2023, with a sustained plateau above 90% since July 2022. There was a significant reduction in the median

**Data availability statement:** All relevant data are within the manuscript and its Supporting information files. This was a quality improvement project. Ethics approval was not obtained to make the data publicly available.

**Funding:** 1. Grand Challenges Canada (grant code: 2008-35944) - The funders had no role in study design, data collection and analysis, decision to publish, or preparation of the manuscript - PI: NK - Salary Contribution: : IA, SK, JK, CA, BO, NK, DD, BH, CZ, CH, 2. Mining4Life https://mining4life.org/ - Co-PI: MA, NK and BC Children's Hospital Foundation - The funders had no role in study design, data collection and analysis, decision to publish, or preparation of the manuscript - Salary Contribution: : IA, SK, JK, CA, BO, NK, DD, CZ, CH, SN 3. Michael Smith Health Research BC Trainee Award (RT-2022-2583) - Trainee salary contributions: YP - The funders had no role in study design, data collection and analysis, decision to publish, or preparation of the manuscript

**Competing interests:** The authors have declared that no competing interests exist.

time-to-antimicrobials during the 5-year period, from 77.6 to 53.6 minutes, with a slope of −0.4 minutes per month (CI: −0.73 to −0.04, p-value: 0.029). The inpatient mortality rate decreased from 5.1% in August 2018 to 2.6% in October 2023, with a significant increase in the number of cases with comparable illness severity.

### Conclusion

The impact of Smart Triage was sustained beyond the end of the feasibility trial and showed sustained improvements in processes such as treatment times and clinical outcomes including a reduction in mortality. HICH's leadership integrated a culture of QI across disciplines and departments, contributing to this initiative's sustainability and impact.

---

## Introduction

Sepsis is one of the leading causes of death among critically ill children, with sub-Saharan African countries reporting the highest mortality rates [1,2]. Effective sepsis care relies on timely identification and treatment [1]. In low-resourced emergency settings, the challenges of identifying and treating sepsis in children are compounded by staffing shortages, low stock of medications, and patient overcrowding [1,3,4].

Triage is a crucial part of emergency care in children. During triage, health workers categorize children based on their presenting symptoms and severity of illness to inform the order in which they are prioritized for clinician assessment and treatment [5,6]. Triage is critical to identify children who are at imminent risk of death and need urgent treatment [7].

Numerous triage processes have been designed for low-resource settings, including the World Health Organization's (WHO) Emergency Triage Assessment and Treatment (ETAT) system [8], the Pediatric South African Triage Scale [9] and the Integrated Management of Childhood Illnesses guidelines [10]. Despite repeated efforts across different countries and regions, effective triage remains rare in low-resource settings and patients are often assessed by clinicians on a first-come-first-served basis, leading to preventable delays [11]. Children are vulnerable to rapid decompensation during a critical illness, leading to a higher risk of death. Thus, there is an urgent need to deploy effective triage systems for children in low-resource settings [8,12].

Contemporary guidelines recommend the initiation of antimicrobial therapy as soon as possible for children with sepsis [1]. Early antimicrobial administration as part of a sepsis bundle which includes oxygen, fluids and glucose, is essential in low-resource settings to forestall deterioration and death and decrease the need for critical care support [4,13–15].

The quality of healthcare is suboptimal in many low and middle-income countries. More than 8 million people per year die from conditions that should be treatable by the health system [16]: nearly 60% of these deaths are preventable but occur due to poor quality of care. Indeed, the mortality burden attributable to suboptimal care is

estimated to be larger than that due to lack of access to care. Hence, quality improvement (QI) efforts are vital, but they must take a holistic approach based on the local context and resource constraints in those settings.

Smart Triage is a digital platform that enables healthcare workers to prioritize children needing urgent or emergency treatment, deploy resources to facilitate timely care and improve the quality of care and clinical outcomes. The platform consists of the Smart Triage mobile application, the Smart Spot Bluetooth Low Energy (BLE) system to track patient location and treatment administration, and a clinical dashboard for highlighting priority patients and viewing triage and treatment data (Fig 1) [17]. Smart Triage provides a triage category using an externally validated predictive model plus independent triggers (danger signs) that are easily collected during an initial assessment [18]. The Smart Triage model [19] predicts admission based on nine variables, including anthropometrics, clinical signs, and vital signs.

## Methods

Ethics approval for this project was obtained from institutional review boards at the Mbarara University of Science and Technology Institutional Review Committee (reference no. 11/02–18), Makerere University School of Public Health (ID: SPH-2021–41), and the Uganda National Council for Science and Technology (ID: HS1745ES). As a QI initiative, ethics approval was not required by the University of British Columbia.

SQUIRE 2.0 Guidelines were utilized to report the findings of this QI initiative [20].

### Setting

*Smart Triage + QI* was implemented at HICH, a private, non-profit, faith-based pediatric hospital accredited by the Uganda Catholic Medical Bureau (UCMB) (S1 Fig). HICH provides subsidized fee-for-service outpatient and inpatient care for children up to 17 years of age. This initiative occurred in the OPD, which provides care for over 25,000 children annually. The OPD is operational 24 hours a day and 7 days a week. At the beginning of this initiative, the OPD consisted of a triage room, a waiting area, three consultation rooms, a laboratory, a pharmacy and one clinical treatment room. All patients presenting at HICH's OPD participated in this QI initiative, except children presenting for vaccinations, elective surgery, or wound dressing changes. The triage platform was customized to communicate with HICH's Electronic Medical Record (EMR) to ensure interoperability between existing health records and minimize the need for duplicate data entry.

### The Smart Triage platform

The Smart Triage mobile application includes pulse oximetry with a connected pulse oximeter sensor attachment (PhoneOx) [18], an integrated respiratory rate counter (RRate) [21], and an externally validated risk prediction algorithm [22,23].

The Smart Spot BLE system consists of color-coded lanyards, based on patients' triage categories, with an attached BLE patient beacon, treatment beacons, and BLE readers. At the end of treatment, the beacon number (labelled on the beacon) is entered with the patient's data and sent to the dashboard. The BLE readers were strategically located in the hospital to provide the patient's location as they moved through the facility by detecting the signal strength from each beacon and communicating with the local server over Wi-Fi. Signal strengths exceeding a prespecified threshold were considered in that room (thresholds were identified based on room size and location in baseline testing). The three triage categories were non-urgent, priority, and emergency, and the respective patients are given green, yellow, and red lanyards to indicate their triage acuity score. Treatment beacons for each treatment type (IV antibiotics, IV fluids, oxygen, nebulization, and antimalarials) were located in the treatment room and are used to easily track the times-to-treatment as the difference between the arrival time and the time when the beacon button was pressed. The nurse administering the treatment pushes both the button on the patient's beacon and the button on the treatment beacon (within 10 seconds) to record a treatment. The Smart Spot system allows accurate and easy data collection for time-to-treatment, supporting a data-driven QI approach. Treatment times can also be manually added to the clinical dashboard.

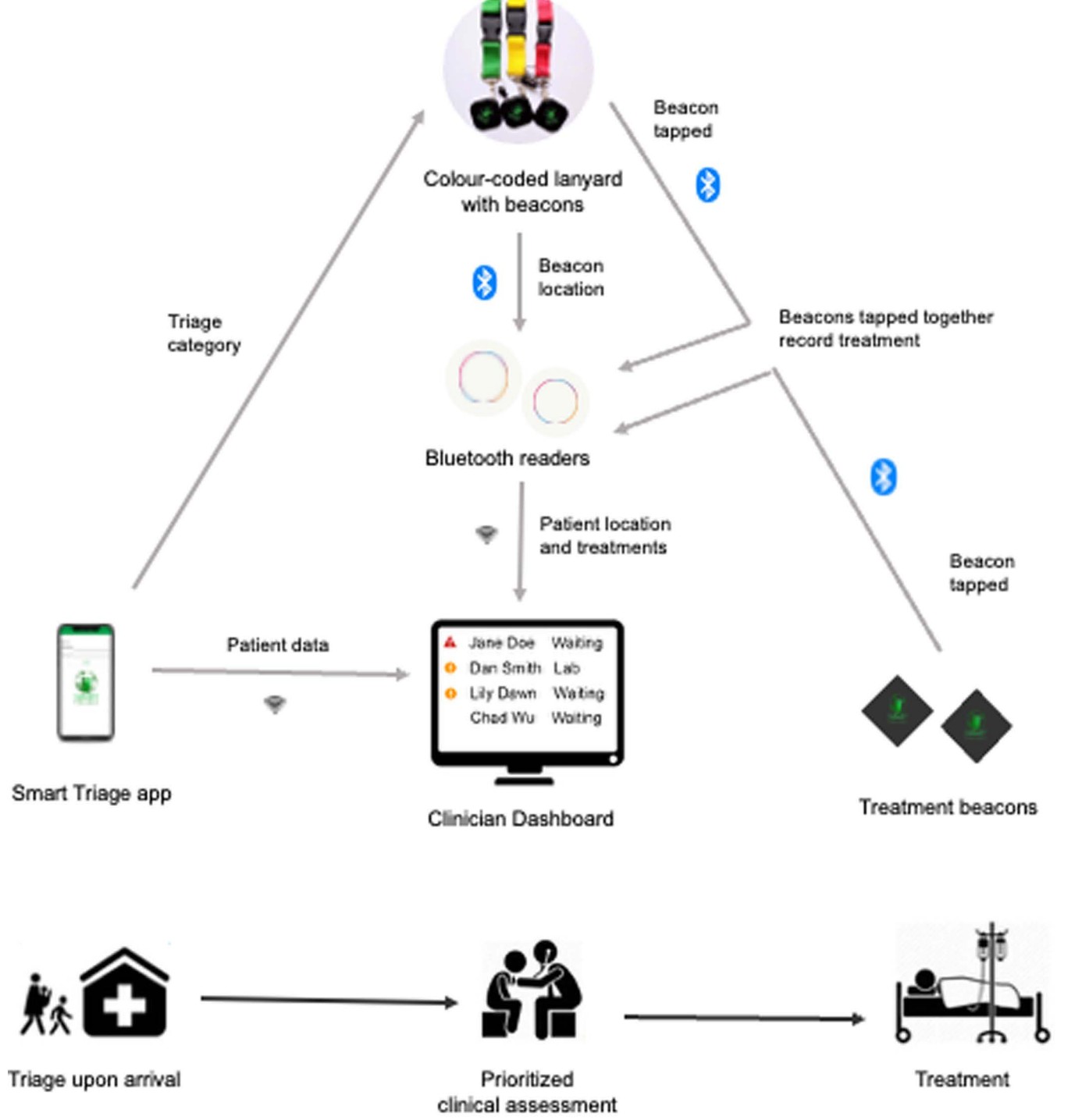

**Fig 1. Smart spot and smart triage platform.**

The Smart Triage clinical dashboard is a password-protected website accessible by trained clinical staff. The dashboard displays patients' triage categories, assessment status, location, and treatment data in real-time on large screens, computers, and tablets. The dashboard and underlying database are hosted locally on a secure server on-site, so patient data is inaccessible off-site. On the dashboard, patients are automatically prioritized for full clinical assessment based on the triage category and wait times. The dashboard produces customized aggregate reports, including triaging rates and times for treatment, which can be viewed by clinical staff and hospital administrators to inform QI initiatives.

### QI interventions

Overall, QI efforts were aimed at increasing the triaging rate to 100% in the OPD and reducing the time-to-antimicrobials to below a median of 45 minutes to reduce the number of patients waiting for longer periods. The HICH QI Team selected these thresholds.

### QI training

The Smart Triage + QI Training Program was designed to teach healthcare professionals to lead continuous data-driven QI to improve quality of care. The initial QI training at HICH was led by a QI implementation team including Canadian and Ugandan team members. An introductory meeting with 29 hospital staff members provided an overview of the Smart Triage platform. More in-depth QI training was delivered to a core interdisciplinary QI committee, which included nine staff members: one physician, six registered nurses, one lab technician, and one records staff member. The training occurred for three hours per day over three days and used a train-the-trainer model [24]. A manual [25] that was adapted by the Smart Triage Team from the Uganda Ministry of Health's QI guide was used to support the comprehensive training and activities [26].

The Smart Triage QI Team identified three key drivers of triaging rates and wait time-to-antimicrobials in the OPD (Fig 2): standardized training processes for OPD staff, OPD workflow efficiency, and QI Team communication. Potential QI interventions to address the key drivers were then prioritized and organized into multiple simultaneous Plan-Do-Study-Act (PDSA) cycles (Fig 3).

### PDSA Cycle #1 – Standardize training

The training processes for OPD staff were standardized by including the clinical staff in the initial QI training. The Smart Spot system was implemented before the Smart Triage mobile application to support the OPD staff's understanding of the beacon technology and treatment tracking. Security personnel were integrated into this PDSA cycle to prevent the loss or misplacement of the beacons and lanyards and ensure their retrieval at the hospital entry/exit gate.

The Smart Triage platform was launched to support triage quality by standardizing triage assessments, identifying the patients most in need of treatment, and reducing the time to complete triage. There was bi-annual triage refresher training for existing OPD staff and regular orientation of new OPD staff (including intern doctors) to the Smart Triage platform. After one year of Smart Triage implementation, the Smart Triage QI Team noted a need for pediatric-specific assessment training for the OPD staff. A workshop overviewing pediatric assessment skills was completed in November 2022 for the OPD nurses and clinical officers. HICH leadership prioritized consistency in their OPD staff evaluations and feedback.

### PDSA Cycle #2 – Adjust workflow

The OPD workflows were adjusted to make triaging and treatment administration more efficient. Communication between Smart Triage and HICH's EMR eliminated the need for duplicate demographics data entry by OPD staff and ensured triage data was stored long-term in the EMR. The Smart Triage clinical dashboard improved the identification of the patients with the highest priority, which enabled the nurses and staff at the reception desk to effectively order patients for their full

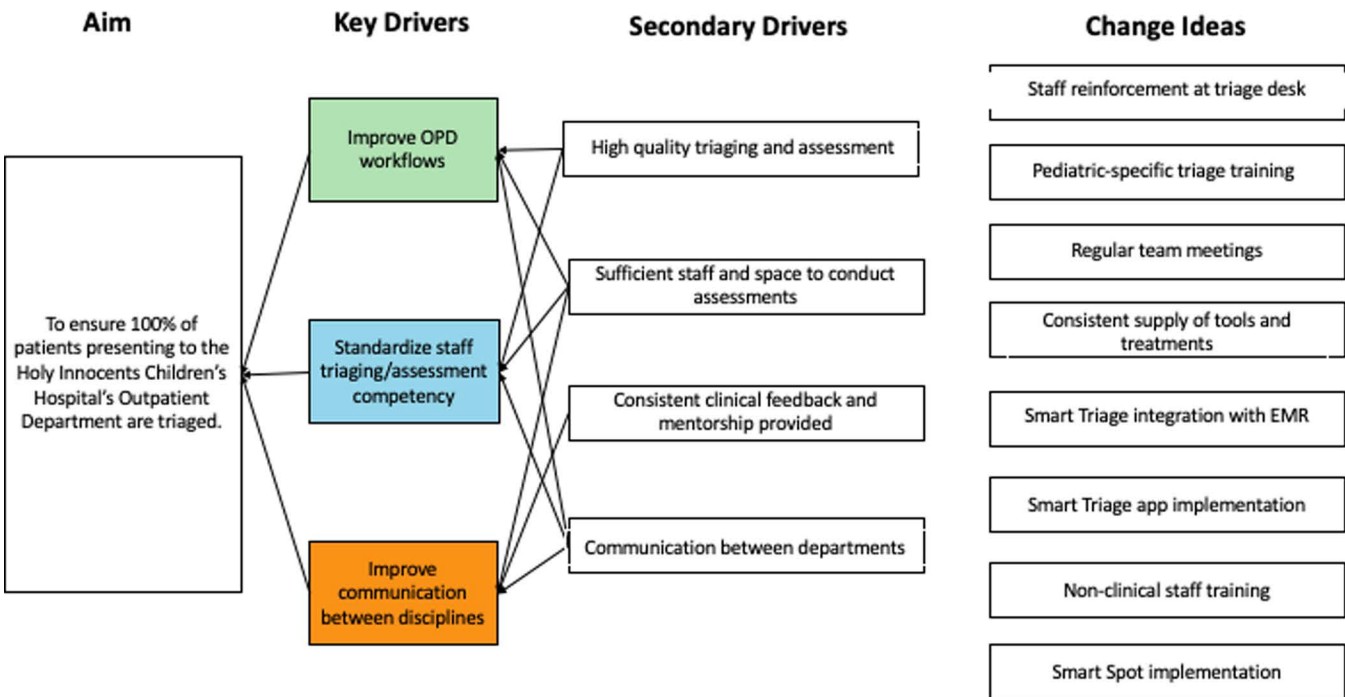

**Fig 2. An example of a key driver diagram to optimize triage rates was developed during the quality improvement process.** The diagram was used to target specific interventions, and their effectiveness was measured during the Plan-Do-Study-Act cycles. Similar diagrams were used for other quality improvement initiatives, such as waiting times in the laboratory or completion of vital sign measurements.

clinical assessments. Changes to the clinical dashboard improved patient flow management by tracking clinician assessment and linking records directly to the patient's OPD EMR to improve patient charting.

The Smart Triage QI Team collaborated with the Pharmacy Department to mandate antimicrobial administration in the OPD prior to the patient admission, rather than on the ward as per the previous workflow. The Smart Triage QI Team partitioned an unused clinical room into an additional consultation and radiology room for imaging. They also ensured the EMR and Smart Triage dashboard were available on computers in every office in the OPD, allowing them to be converted into consultation rooms if patient volume required additional space. The treatment room was also moved to a larger room with more cribs for children. HICH leadership supported this change by increasing their nursing staff from two nurses to four nurses per shift and adding two more physicians to the OPD as the volume of patients increased.

### PDSA Cycle #3 – QI Team communication

QI Team communication was reorganized by creating two different teams: an HICH QI Committee and Smart Triage QI team. The HICH QI committee was responsible for hospital-wide QI initiatives while the Smart Triage QI Team championed Smart Triage within the OPD and led the PDSA interventions. To prioritize the Smart Triage QI initiative, a Smart Triage QI Project Lead at HICH was identified and partnered with the pre-existing HICH QI Committee physician lead. The Smart Triage QI Project Lead attended the monthly hospital-wide QI meetings focused on hospital-wide QI initiatives to promote Smart Triage.

The Smart Triage QI Team initiated OPD-specific weekly QI meetings to discuss the results of the weekly data reports. They utilized WhatsApp as a frequent team communication channel. The initial QI training for OPD staff was

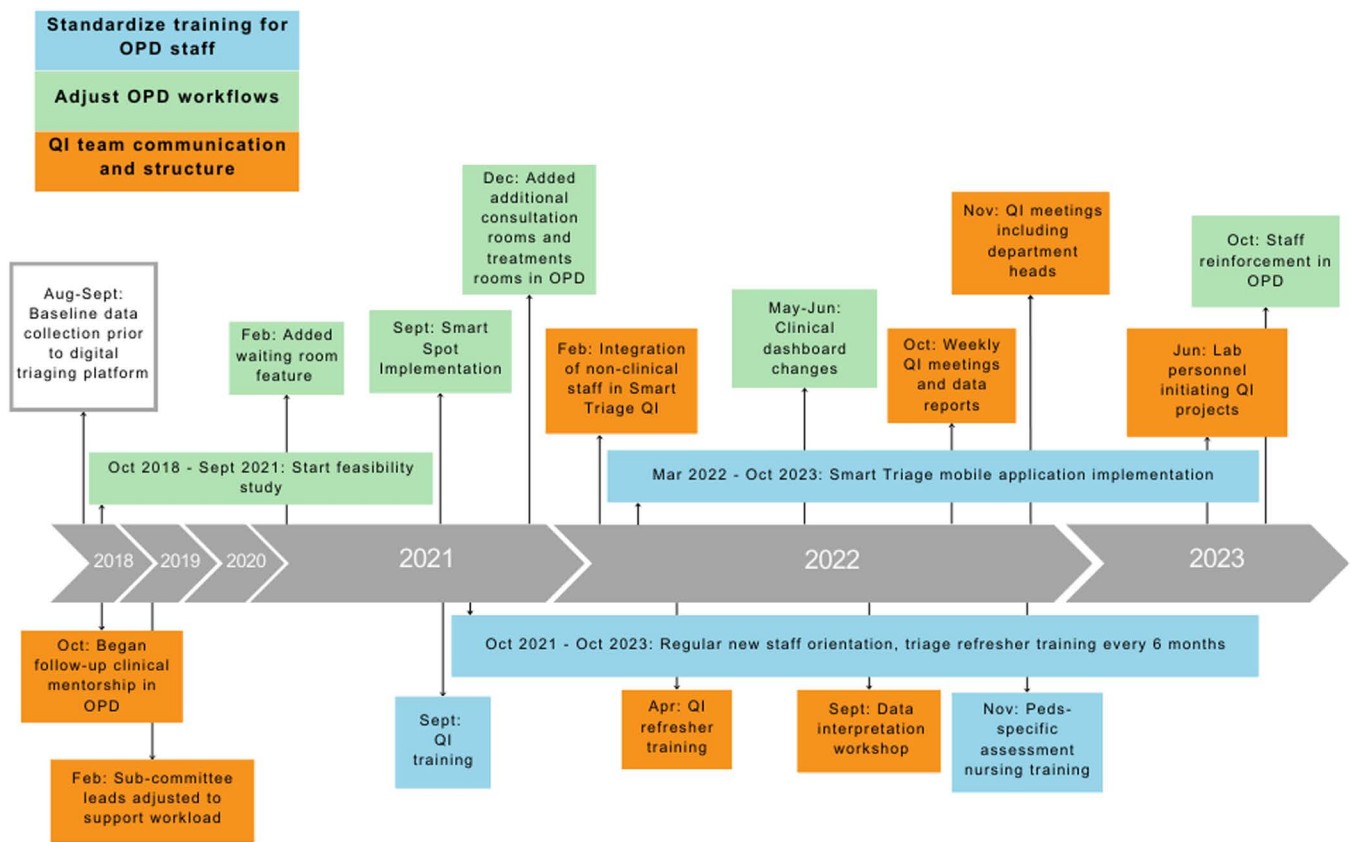

**Fig 3. HICH Smart Triage QI Interventions Timeline October 2018 – October 2023.** The platform feasibility study was conducted between 2018 and 2021, and a formal quality improvement study was conducted from 2021 to 2023.

consistently reinforced by QI refresher workshops led by the HICH QI Team leaders. De-identified data from Smart Triage were summarized, and key metrics were reported weekly to the HICH QI Team, hospital administration, and QI leads at the UCMB via email and to a broader group of clinicians at HICH via WhatsApp Messenger. Reports included summary data from the triages that week, including patients per day, patient age and sex, top five presenting complaints that week, total triages and admissions in each triage category, total treatments administered in each treatment category, time-to-first treatment, pulse oximetry quality, and run charts [27] of the median time to IV antimicrobials and number of triages per week. The HICH QI project lead also conducted data interpretation training with the OPD staff so they could interpret the data received in the weekly reports. This strengthened internal efforts to maintain a data-driven QI project.

A Smart Triage implementation trial was ongoing at four other sites in Uganda and Kenya; therefore, virtual meetings were held weekly with project leads from each site and Smart Triage study team members in Canada. Initial meetings focused on technology implementation, site progress updates, and ongoing challenges within the PDSA cycles. This provided ongoing support between sites and led to creative strategies to address challenges.

## Data sources

For this analysis, data was collected from HICH's monthly reports (accessed on January 11, 2024, aggregate data), the Smart Triage platform (accessed on November 30, 2023, deidentified data) and the EMR (open-source platform Care2X;

used to obtain OPD attendance, accessed on December 7, 2023, deidentified data). HICH is mandated to provide monthly reports to the Ugandan Ministry of Health, including reports of mortality rates and admission rates.

## Evaluation and analysis

HICH's primary outcomes were the triaging rates within the OPD, the median time from triage to initiation of antimicrobials (minutes), and the mortality rates of admitted patients from the monthly facility mortality reports at HICH (mortality data for triaged patients was only determined during the study periods). Therefore, we evaluated the change in five variables (OPD attendance, acuity of illness, triaging rates, median-time-to-antimicrobials, and mortality rates of admitted patients) over the duration of the intervention using run charts with monthly data points. We began collecting baseline data with the initial launch of the feasibility study in August 2018; therefore, we have included this data to establish a strong baseline for our data analysis. However, triaging rates and acuity of disease had a start date of November 1, 2018, the start of the first full month with the triage platform for the feasibility study. The end date of October 31, 2023 was the same for all variables (Table 1).

The acuity of illness was studied to determine whether there was any significant change in illness severity among the patient population during the 5-year period. We retrospectively assessed illness severity using the Emergency Department-Pediatric Early Warning Score (ED-PEWS) [28,29]. ED-PEWS relies on seven vital signs plus child's age to classify children into three categories, which are high, medium, and low urgency [29]. Triaging rates were calculated by dividing monthly triages by the monthly OPD encounters from the EMR. Prior to mid-Oct 2018, triage and treatment times (used to calculate the monthly median-time-to-antimicrobials) were manually recorded on paper for the feasibility study [30]. After implementation, these variables were recorded in the feasibility study and, subsequently, Smart Triage. Mortality rates were calculated by dividing the number of monthly deaths by the number of admitted patients, with no exclusions. Deaths and admissions were accessed from HICH's monthly reports to the Ministry of Health.

The overall trajectories of change of OPD attendance, triaging rates, illness acuity (independent ED-PEWS urgency categories), time-to-antimicrobials, and mortality rates from the facility monthly statistics of all patients were assessed using simple linear regression with time as the explanatory variable. A variable was considered to significantly change over time if the 95% CI of the slope of the linear regression line did not include zero. Linearity was assessed using the rainbow test. Missing data points on the run charts were linearly interpolated. To correct the p-values for multiple comparisons, we used a False Discovery Rate adjustment [31]. We chose a significance level of 0.05 for all tests conducted. We analyzed data using Microsoft Excel (Version 16.86) and the "stats" package in R software (version 4.3.1) [28,32].

**Table 1. Description of data sources and variables collected. The platform feasibility study was conducted between 2018 and 2021 and a formal quality improvement study was conducted from 2021 to 2023.**

| Data source | Dates available | Type of data | Source variables |
|---|---|---|---|
| Monthly reports | Aug 2018 – Oct 2023 | Reports generated by the hospital for the Ugandan Ministry of Health. | • Admission number per month<br>• OPD Attendance number per month<br>• Mortality number per month |
| Research data platform and the Digital Triaging platform | Aug 2018 – Oct 2023 | Collected by the research team during the baseline phase and recorded by the triage application for feasibility and quality improvement studies. | • Patient demographics<br>• Number of triages (numerator)<br>• Vital signs required for ED-PEWS illness acuity.<br>• Time to administration of antimicrobials |
| Electronic medical record | Aug 2018 – Oct 2023 | Care2X open-source application used in the outpatient department. | • Number of triages (denominator) |

## Results

From August 1, 2018, to October 31, 2023, 121,521 children attended the OPD at HICH. November 2018 was the first full month of triaging children using our digital triage platform. The median (IQR) age was 23.9 (8.0 to 55.9) months (Table 2). OPD attendance increased significantly over time with a slope of simple linear regression of 18 patients per month (95% CI: 7.0 to 26.9, p-value <0.05) (Fig 4). There is no evidence of non-linearity (F = 1.0, p = 0.47).

In this cohort, 108,957 (89.7%) children had sufficient data for ED-PEWS calculation. The number of high-urgency patients increased by 6.2 patients per month (95% CI: 3.37 to 8.94, p-value <0.01), with a concomitant increase in patients at an overall average of 19% (95% CI: 18–20 and no significant change in the percent of patients that were classified as high-urgency (Fig 5).

The OPD triaging rate increased to 91% by October 2023, with a sustained plateau of 90% since July 2022 (Fig 6). Triaging rates and attendance were reduced during the first COVID-19 lockdown period. The triage rates were already high in 2018 due to the support from the study team at the initiation of the baseline period of the feasibility study.

There were 3,810 antimicrobial treatments documented in the Smart Triage platform between August 1, 2018, to October 31, 2023. Two months of median time-to-antimicrobial administration data (July and September of 2020) were missing (not reported) and linearly interpolated. There was a significant reduction in the median time-to-antimicrobials during the five-year intervention, from 77.6 to 53.6 minutes, with a slope of simple linear regression of −0.4 minutes (−24 seconds) per month (95% CI: −0.73 to −0.04, p-value <0.05) (Fig 7). There was no evidence of non-linearity (F = 0.71, P = 0.83).

There were 883 deaths during the five-year intervention. The mortality rate in admitted patients decreased from 5.1% in August 2018 to 2.6% in October 2023 (49% decrease) (Fig 8). The line of linear regression for the rate of mortality had a slope (95% CI) of −0.04% per month (−0.068% to −0.02%. p-value <0.01). There was no evidence of non-linearity (F = 0.90, p = 0.61).

**Table 2. HICH patient demographics during QI intervention.**

| August 2018 – October 2023 HICH Patient Demographics | | |
|---|---|---|
| **Age (months)** | | |
| Median | 23.9 | |
| IQR | 8-55.9 | |
| **Sex** | **N** | **%** |
| Female | 55,074 | 45.3 |
| Male | 66,446 | 54.7 |
| **Nutrition Status** | | |
| Overweight | 3,324 | 2.7 |
| Normal | 98,834 | 81.3 |
| Moderate Acute Malnutrition | 4,512 | 3.8 |
| Severe Acute Malnutrition without Edema | 8,834 | 7.3 |
| Severe Acute Malnutrition with Edema | 19 | .02 |
| Patients missing data to calculate nutrition | 5,998 | 5.9 |
| **Top 3 Presenting Complaints** | | |
| Fever | 31,750 | 29.1 |
| Cough | 23,927 | 22.0 |
| Flu | 15,093 | 13.8 |

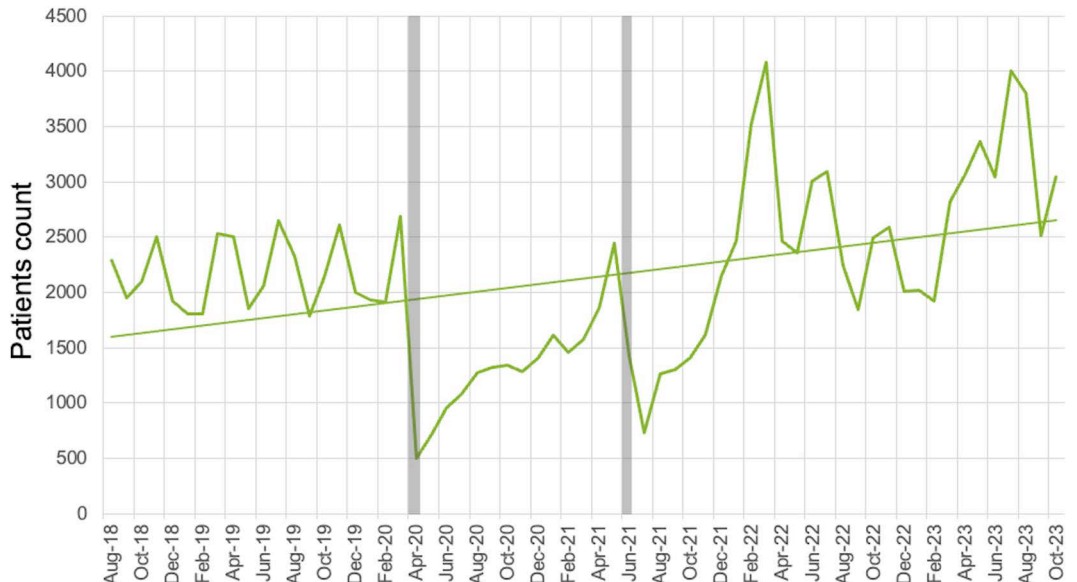

**Fig 4. Number of patients presenting to the OPD per month; COVID-19 Lockdowns indicated by grey bars.**

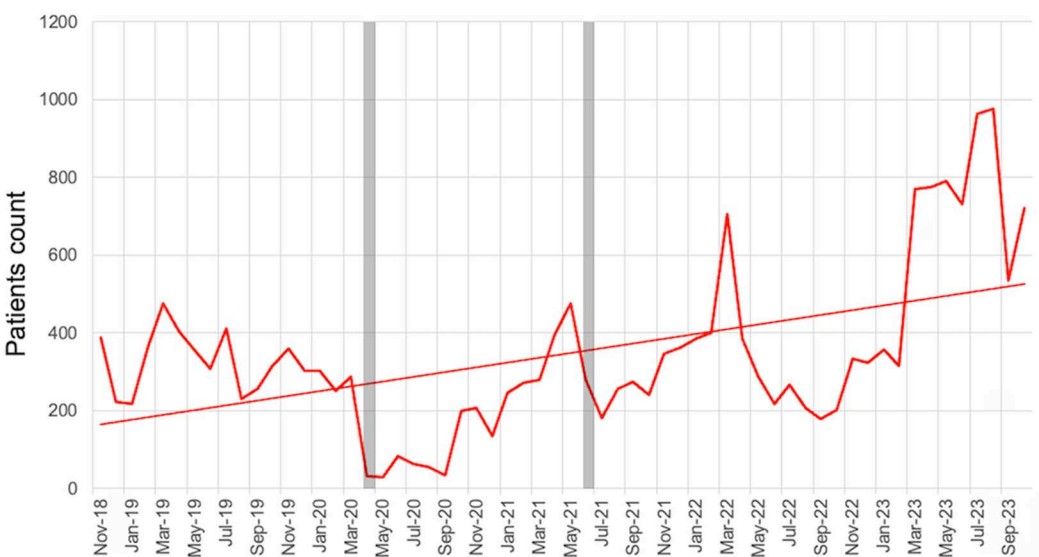

**Fig 5. Monthly count of high urgency patients based on ED-PEWS scores; COVID-19 Lockdowns indicated by grey bars.**

## Discussion

This QI initiative was initiated to sustain triaging rates and reduce time-to-antimicrobials in the OPD of a pediatric hospital in Uganda, which demonstrated significant changes to both process and clinical outcomes. Multiple simultaneous PDSA cycles were initiated and retrospectively organized into three themes: standardize training, adjust workflow, and improve QI Team communication. The initiative resulted in high triaging rates and a sustained reduction in time-to-antimicrobials. Mortality of admitted patients decreased by almost half during this 5-year period, despite a sustained increase in patient

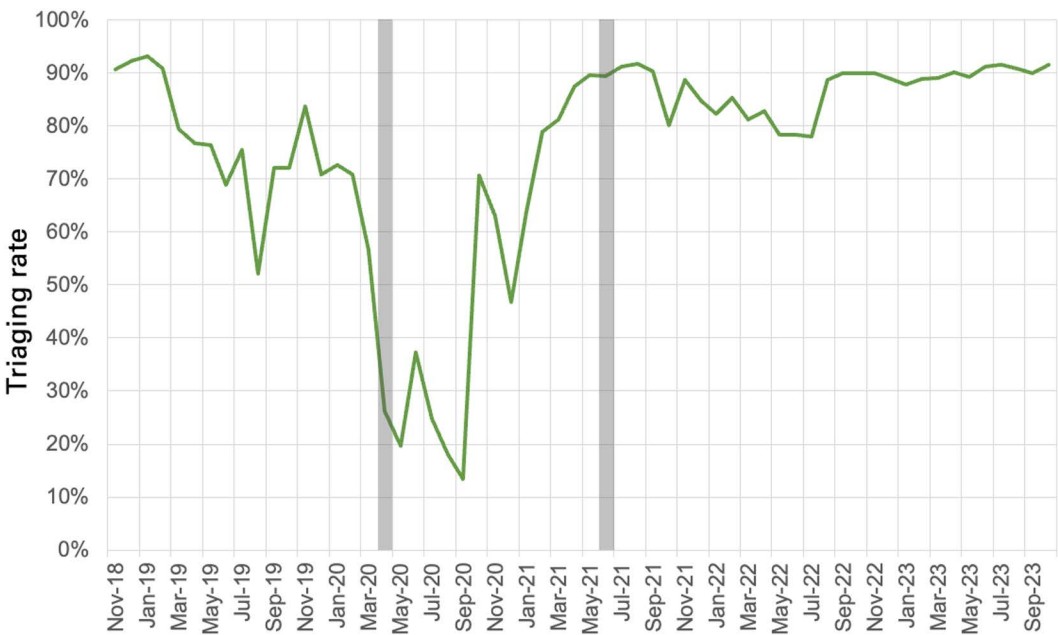

**Fig 6. Triaging Rate in the HICH's Outpatient Department; COVID-19 Lockdowns indicated by grey bars.**

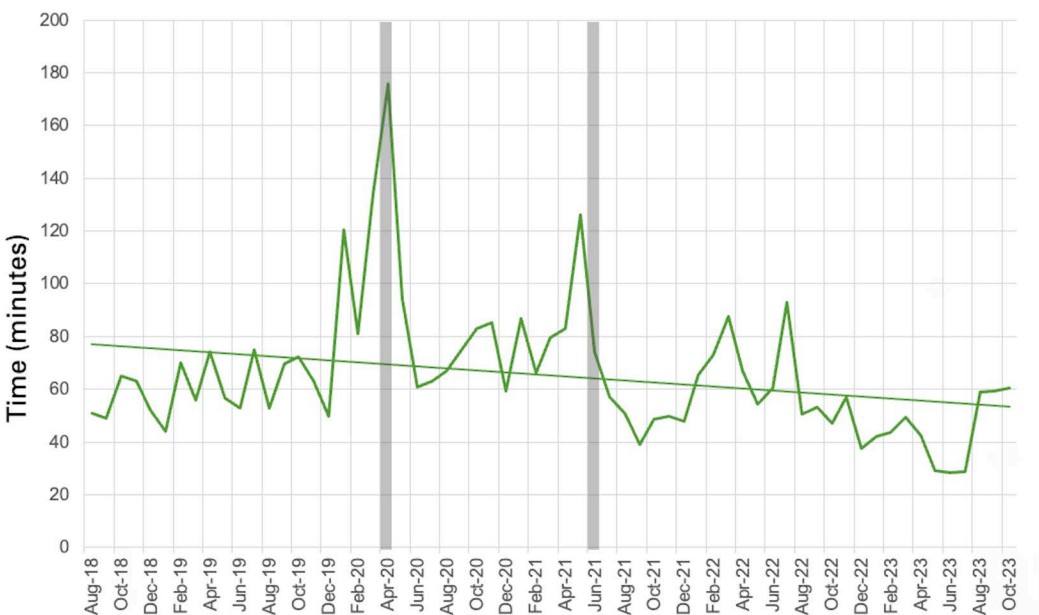

**Fig 7. Median time to receiving antimicrobials at HICH in the OPD; COVID-19 Lockdowns indicated by grey bars.**

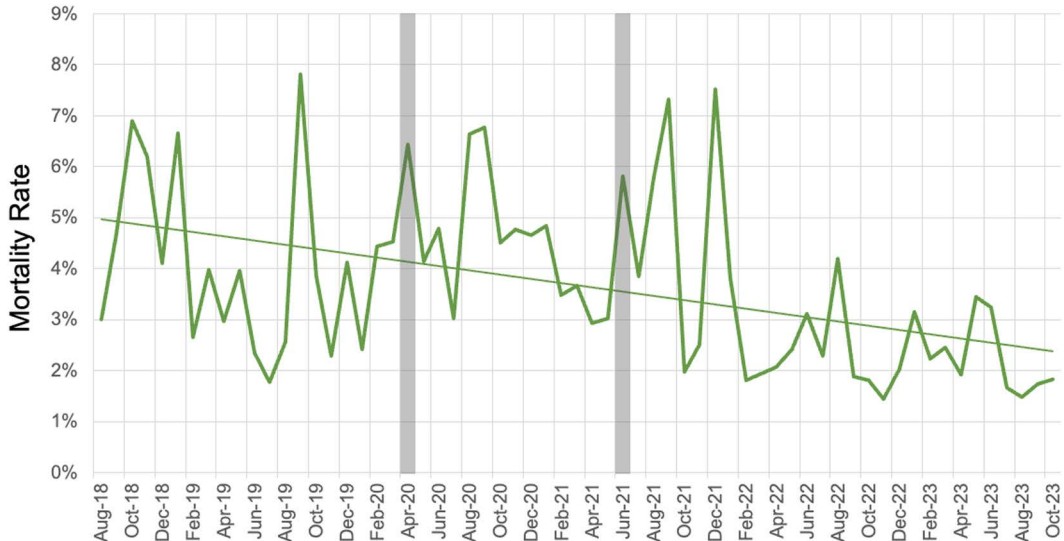

**Fig 8. Mortality Rates of Admitted Patients at HICH; COVID-19 Lockdowns indicated by grey bars.**

numbers with comparable illness severity. However, other factors, such as vaccination rates, case mix, and other facility-wide quality improvements, may have contributed to this mortality reduction.

Process improvements were the focus of this QI intervention; however, the ultimate purpose of all QI efforts was the improvement of clinical outcomes. Implementing a triaging system in low-resource settings has been previously shown to improve inpatient mortality rates [7,33–35]. While decreased mortality rates due to sepsis have been linked to a decreased time-to-antimicrobials [13,14], this is likely also related to changes in clinical awareness and other elements of the sepsis bundle of care, such as intravenous fluid resuscitation [33]. At the start of the intervention, the time-to-antimicrobials at HICH was just above the current Surviving Sepsis Guidelines of one hour [1]. Furthermore, the time-to-antimicrobials significantly reduced to levels that would be highly acceptable in any clinical setting. This change is likely to be highly correlated to other simultaneous QI initiatives. For example, HICH prioritized the education of their nursing staff during this intervention timeframe and supported nurses in receiving diploma-level training. HICH also mandated the consistent supply of medicines on their site, thereby eliminating treatment delays that were due to stockouts.

A stable triage rate above 90 percent is indicative of staff adoption and adherence to the QI process. The remaining 10 percent of patients were mostly those requiring immediate emergency resuscitation and did not need to be triaged using the Smart Triage mobile application or those who arrived during the night when triage may not have been reliably performed. The increased volume of patients is likely due to the increased capacity and the efficient and effective service provided, as well as increased utilization by the community. The QI interventions were sustained even with increasing OPD attendance. This may be attributable to an increase in funding at HICH to support additional staffing, expand infrastructure and improve workflow to accommodate the increased demands.

## The importance of effective triage

Effective triage can be transformative to patient flow and outcomes, particularly in low-resource settings [6,36–38]. Similar projects have been implemented in various low-resource settings to improve pediatric triage [38–40], many with a focus on implementing the WHO's ETAT training [39–41]. ETAT uses clinical emergency signs to categorize patients' priority categories and relies on intensive and consistent training of the triaging personnel to be effective [41–43]. ETAT also includes integrated

management guidelines [42,43] that are not included in Smart Triage. Use of the nine-variable Smart Triage algorithm is rapid (variables can be taken by a trained nurse in less than 5 minutes per child), requires minimal training or experience, and includes the ETAT danger and priority triggers [44]. Furthermore, Smart Triage's risk prediction model with independent triggers has demonstrated higher specificity in identifying emergency patients at risk of death compared to ETAT [44].

### Antimicrobial administration

Ideally, antimicrobial agents should be administered to those who are likely to benefit and withheld from those who do not need it. However, identification of groups likely to benefit is difficult in the absence of microbial and laboratory support. Lack of laboratory support also hampers antimicrobial stewardship programs. That there was a decrease in mortality rate despite less antimicrobial use is heartening, but difficult to explain. Time-to-antibiotics in children with septic shock is associated with decreased mortality and length of hospital stay in high-income countries [15,45–47]. However, it is unlikely that the decrease in time to antibiotics of 24 minutes is solely responsible for better outcomes; better triage and attention to overall care may be responsible, but difficult to prove. The interactions between multifaceted changes in care (often called bundles of care) in this intervention are complex, and optimal outcomes are achieved by adopting the entire bundle. Assigning relative value to individual components is challenging because it is difficult to assess the contribution of each element of the bundle separately. The impact of Smart Triage on prescribing practices and outcomes requires further study.

### QI can lead to sustainable change

Many QI initiatives are criticized because QI initiatives are often completed in siloes and do not lead to sustainable organizational-level change [48]. A key part of strengthening health service delivery is the integration of QI into "the fabric of care itself" [49]. Continuous QI (CQI) embeds itself into institutional culture in healthcare by fostering "a culture of continuous learning, innovation, and improvement" [50].

This QI initiative evaluates HICH's commitment to CQI, which led to sustainable change and impacted long-term health outcomes. Weekly feedback to the clinical team using run charts was effective in easily visualizing progress, required minimal mathematical complexity, and allowed the QI Team to objectively understand if the changes made to the process or system over time led to improvements [51]. HICH's QI leadership was essential for the process and clinical improvements in this initiative and the ultimate sustainability of Smart Triage. HICH's Smart Triage QI Team intentionally integrated QI concepts across many disciplines and departments, which catalyzed a shift toward QI within the entire hospital. The HICH QI Team included department heads in their quarterly QI meetings, which initiated various other departmental QI projects. Lab personnel also began learning QI strategies and have initiated lab-specific QI projects with the Smart Triage QI Team's guidance. The team also began including non-clinical staff in QI training. For example, security guards were incorporated into part of the Smart Spot implementation by tracking beacons. After receiving QI training, members of the security team also indicated a desire to create their own QI projects within the OPD.

### Digital health implications

HICH's commitment to digital innovation and technological advancement is a distinct contributor to its success with the implementation of Smart Triage. The results of this intervention also demonstrate the potential for risk-stratified digital health interventions to catalyze QI in a low-resource emergency department setting. However, like QI initiatives, digital health projects can also occur in siloes and not lead to organizational-level change [52]. Digital health in African settings has historically been limited by a lack of adaptation to different contextual needs and institutional capacities [53]. The Smart Triage algorithm and training material is open source and incudes a Digital Adaption Kit that will allow it to be easily integrated into new EMR systems, as seen at HICH and other hospitals in Uganda, demonstrating future scalability [54]. Digital health can harness existing health resources and extend them further, moving closer toward quality universal health coverage [53,55,56].

### Challenges in implementation

The use of the BLE tracking system, Smart Spot, was an ongoing challenge to the success of this project. Health-care workers and patients communicated concerns regarding the tracking of patient locations and concerns about the health effects of wireless energy waves. Multiple QI initiatives were aimed at reducing beacon loss, which occurred at the expense of other potential initiatives. Treatment tracking using the beacons was difficult to maintain and beacons were eventually phased out of HICH's workflow with the implementation of a new electronic charting system. While the automatic tracking of treatments is desirable and wireless technologies are widely used for tracking in many applications, the barriers to adoption in this context were significant and were not reliably overcome. Smart Triage can be fully implemented without the Smart Spot system by tracking time using the electronic charting system.

Smart Triage relies on the use of a phone or tablet with connected pulse oximeter for optimal functionality. Any loss or malfunction of the pulse oximeter impacts data quality and accuracy. Vital signs can be entered from other available devices; however, this may degrade reliability [57]. Equipment maintenance and replacement due to theft and wear-and-tear must be considered to ensure sustainability. Disruptions to the local network also cause issues with uploading patient data and using the clinical dashboard, but the platform is not reliant on an outside Internet connection which can more frequently be unstable.

The activation and maintenance of a strong hospital QI leadership team was initially a challenge at HICH. Due to time and human resource constraints, monthly QI meetings were often postponed. However, once buy-in was obtained from HICH leadership, they were one of this initiative's main drivers of success. Documentation within the team of the various PDSA cycles was another serious limitation. Many of the dates of the PDSA interventions were based on staff recall due to challenges with prioritizing robust documentation during the initiative.

### Limitations

The major limitation of this study is that multiple data sources were used, each with their own shortcomings. We chose to use the monthly hospital mortality reports as our study data was focused on triage and treatment tracking, rather than mortality, and access to the EMR was limited to patient counts. Not all patients were recruited to the study and the observation period covered more than one study. The mortality reduction observed was likely not due to the Smart Triage intervention alone. There may have been sustained improvements in early resuscitation and other areas such as radiology, laboratory, and pharmacy beyond the research period. In addition, it is possible that other interventions, such as improved vaccination rates, improved healthcare seeking or improvements in different areas of the facility, may account for the sustained mortality reduction.

### Conclusion

We observed a sustained improvement in triaging rates and time-to-antimicrobials in HICH's OPD through multiple simultaneous PDSA cycles that targeted OPD staff training and OPD workflows while fostering a QI culture within the OPD. The decrease in mortality rates for admitted patients at HICH suggests the real impact of this initiative. Still, we cannot rule out other changes in the delivery of health services as contributors. The success of digital health in low-resource settings relies on the consistent commitment of a leadership team and the continuous adaptation to its environment; this supports the continuous improvement of quality of care and the goal of universal health coverage.

### Supporting information

**S1 Fig. Organizational Chart for Holy Innocents Children's Hospital.**
(TIFF)

## Acknowledgments

We are grateful for the quality improvement efforts of facility staff and leadership at Holy Innocents Children's Hospital. We would like to acknowledge the persistent efforts of the outpatient department nurses including and not limited to Rita Ahebwa, Milia Ampeire, Alex Aruho, Bannet Gumisiriza, Matthias Mugabe, Jonan Nuwagaba, Tadius Singura, Winnie Tugume, Darius Tugumenawe, Phionah Turyatemba and Cleophus Twongirwe. We would also like to thank the team of physicians at HICH, including but not limited to Dr. Hillary Asiimwe, Dr. Ezra Kushaba, Dr. Paul Mulyamboga, Dr. John Mzee, Dr. Marvin Luzinda Wakubirwa, Dr. Kenneth Wandera and Dr. Felix Yetungye. Further, we would like to thank members of the Smart Triage research team based at Walimu, Uganda; and the Institute for Global Health, Canada. This includes but is not limited to Collins Agaba, Emmanual Bamwesigye, Clare Komugisha, Katija Pallot, and Jessica Rigg.

## Author contributions

**Conceptualization:** Yashodani Pillay, Dustin Dunsmuir, Charly Huxford, Niranjan Kissoon, J. Mark Ansermino.

**Data curation:** Dustin Dunsmuir.

**Formal analysis:** Ahmad Asdo, Dustin Dunsmuir.

**Funding acquisition:** Stefanie K. Novakowski.

**Investigation:** Yashodani Pillay, James Karugaba, Stephen Businge, Mike Kyewalyanga.

**Methodology:** Yashodani Pillay, Ivan Aine Aye Ishebukara, Dustin Dunsmuir, Charly Huxford, Fredson Tusingwire, J. Mark Ansermino.

**Project administration:** Yashodani Pillay, James Karugaba, Ivan Aine Aye Ishebukara, Dustin Dunsmuir, Justine Behan, Charly Huxford, Stefanie K. Novakowski, Fredson Tusingwire, Gloria Kakuru, J. Mark Ansermino.

**Resources:** Yashodani Pillay.

**Software:** Ivan Aine Aye Ishebukara, Dustin Dunsmuir.

**Supervision:** Yashodani Pillay, James Karugaba, Ronald Kasyaba, John Khisa, Stephen Businge, Mike Kyewalyanga, J. Mark Ansermino.

**Validation:** Ahmad Asdo.

**Visualization:** Ahmad Asdo, Dustin Dunsmuir, Charly Huxford.

**Writing – original draft:** Rebecca Goertzen, Ahmad Asdo, Justine Behan.

**Writing – review & editing:** Yashodani Pillay, James Karugaba, Ahmad Asdo, Dustin Dunsmuir, Charly Huxford, Stefanie K. Novakowski, Fredson Tusingwire, Ronald Kasyaba, Gloria Kakuru, John Khisa, Stephen Businge, Mike Kyewalyanga, Niranjan Kissoon, J. Mark Ansermino.

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
