## [Decision Letter · Decision Letter 0]

PONE-D-24-57615Improving pediatric care in Uganda with a digital platform and quality improvement initiative: a retrospective review of Smart Triage + QIPLOS ONE

Dear Dr. Goertzen,

Thank you for submitting your manuscript to PLOS ONE. After careful consideration, we feel that it has merit but does not fully meet PLOS ONE’s publication criteria as it currently stands. Therefore, we invite you to submit a revised version of the manuscript that addresses the points raised during the review process.

We look forward to receiving your revised manuscript.

Kind regards,

Tai-Heng Chen, M.D., Ph.D.

Academic Editor

PLOS ONE

Reviewers' comments:

Reviewer's Responses to Questions

**Comments to the Author**

1. Is the manuscript technically sound, and do the data support the conclusions?

Reviewer #1: Yes

Reviewer #2: Partly

2. Has the statistical analysis been performed appropriately and rigorously? 

Reviewer #1: Yes

Reviewer #2: No

3. Have the authors made all data underlying the findings in their manuscript fully available?

Reviewer #1: Yes

Reviewer #2: Yes

4. Is the manuscript presented in an intelligible fashion and written in standard English?

Reviewer #1: Yes

Reviewer #2: Yes

5. Review Comments to the Author

Reviewer #1: Title: Improving pediatric care in Uganda with a digital platform and quality improvement initiative: a retrospective review of Smart Triage + QI.

General comment: The smart Triage and QI were implemented in the OPD. The results on mortality reduction observed in inpatients may not not necessarily be accounted for solely by interventions in the OPD.

Abstract:

Ordinarily discussion is not acccommodated in the abstract instead a conclusion is more appropriate.

Introduction:

Smart Triage is mentioned briefly wheraas QI is ommitted in the introduction.

Results:

Figures ought to have legends/titles on the bottom.

Discussion:

Whereas there was an observed reduction of mortality during the review perod my concern is to link it directly to the interventions in OPD unless if the QI was hospitalwide. The Authors should first focus on traige rates and time to antimicrobials and can only state that the interventions could have contributed to the reductions in mortality seen.

Conclusion:

Authors are expected to write concise statements in the conclusion section. References are not expected in this section.

Reviewer #2: The study was focused on the feasibility and implementation of the Smart Triage, within the Quality Improvement initiative in the outpatient department (OPD) of a pediatric hospital in Uganda, over a 5-year period (2018-2023). Smart Triage joins a risk prediction algorithm and a digital platform to enable healthcare workers to triage patients and track treatments effectively.

The authors stated that “the aim of QI initiative was to improve triaging rates and to reduce time-to antimicrobials in the hospital’s outpatient department (OPD)”. Moreover, in the section “QI intervention” they declared: “QI efforts were aimed at increasing the triaging rate to 100% in the OPD and reducing the time-to-antimicrobials to below a median of 45 minutes to reduce the number of patients waiting for longer periods”. Finally, at the end of the “Introduction”, the authors affirm: “This paper reports the most significant system-level changes that contributed to sustained improvements in these metrics”.

The study is interesting, but there are some critical issues that would be important to clarify.

• Smart Triage is described in the Introduction (lines 69-77) and in the Smart Triage Platform section (lines 107-129) as a digital platform that consists of three components: 1) a smart triage mobile application; 2) a smart spot bluetooth low energy system; 3) a clinical dashboard. All the Smart Triage system is summarized in figure 1. However, can the authors better clarify how exactly beacons work?

• In the figure 2 “Key Driver Diagram for HICH” is shown the diagram summarizing the Smart Triage QI Team identification of three key drivers of triaging rates and wait time-to antimicrobials in the OPD. However, in the box “Aim” only the triaging rate objective is reported. Furthermore, the conceptual and graphical relationships between aim, key drivers, secondary drivers, and change ideas, should be better explained.

• In the figure 3 “HICH Smart Triage QI Interventions Timeline October 2018 - October 2023” is shown the time schedule of the activities implemented. It would appear that the implementation of the smart triage mobile application took place between March 2022 and October 2023. However, further on in the manuscript (lines 228-229) it says: “November 1, 2018, the start of the first full month with the triage platform”. Since the smart triage mobile application would seem indispensable to the triage platform, would it be possible to clarify this apparent time contradiction?

• A more relevant issue is about “Data sources”. The authors affirm (lines 214-218): “For this analysis, data was collected from HICH’s monthly reports (aggregate data), the Smart Triage platform (deidentified data) and the Electronic Medical Records (deidentified data). HICH is mandated to provide monthly reports to the Ugandan Ministry of Health, including reports of mortality rates and admission rates”.

Can the authors clarify what deidentified data are? If they were individual data, why were all analyses instead conducted on aggregate data derived from the hospital's monthly reports? Was it conceivable to construct an individual-based dataset from the Electronic Medical Records, which would have allowed analytically, not ecologically, to assess associations between certain risk factors and the outcomes studied?

Evaluation and analysis

• Lines 220-222: “HICH’s primary outcomes were the triaging rates within the OPD, the median time from triage to initiation of antimicrobials (minutes), and the mortality rates of admitted patients at HICH”.

Why was the evaluation of mortality rates not stated early on as one of the objectives of the study?

• Lines 224-226: “We began collecting baseline data with the initial launch of PocketDoc in August 2018; therefore, we have included this data when possible, to establish a strong baseline for our data analysis”.

What do the authors mean by “therefore, we have included this data when possible”? What data might not have been included?

• Lines 243-245: The overall trajectories of change of OPD attendance, triaging rates, illness acuity (independent ED-PEWS urgency categories), time-to-antimicrobials, and mortality rates were assessed using simple linear regression with time as the explanatory variable”.

Has the linearity of the outcomes over time been assessed?

Results

• Are the results in Table 1 derived from data already aggregated monthly? In any case, why did the authors not use the variables shown in the table to calculate specific rates (for age, sex, nutrition status, presenting complaints)?

• Lines 259-261: “The number of high-urgency patients increased by 6.2 patients per month (95% CI: 3.37 to 8.94, p-value <0.01); keeping pace with the increase in patients at an overall average of 19% (95% CI: 18-20) of total patients (no significant change in the percent).

• Lines 271-273: There was a significant reduction in the median time-to-antimicrobials during the five-year intervention, from 77.6 to 53.6 minutes, with a slope of simple linear regression of -0.4 minutes (-24 seconds) per month (95% CI: -0.73 to -0.04, p- value <0.05)”.

It would be helpful if graphs similar to those in Figures 4 and 5 were also produced for these three outcomes, to enable the reader to assess trends over time.

• Lines 261-263: “The OPD triaging rate increased to 91% by October 2023, with a sustained plateau above 90% since July 2022 The OPD triaging rate increased to 91% by October 2023, with a sustained -plateau of 90% since July 2022 (Fig 4)”.

The triage rate at the end of the observation period is broadly similar to that in 2018, before the implementation of the Smart Triage platform.

• Lines 274-277: “The mortality rate in admitted patients decreased from 5.1% in August 2018 to 2.6% in October 2023 (49% decrease) (Fig 5). The line of linear regression for the rate of mortality had a slope (95% CI) of -0.04% per month 277 (-0.068% to -0.02%. p-value <0.01)”.

The mortality trend over time would appear to be nonlinear. Moreover, as mentioned above, it would be interesting to calculate specific rates for the available risk factors. Why has it not been done?

Discussion

• Lines 283-288: “This QI initiative was initiated to sustain triaging rates and reduce time-to-antimicrobials in the OPD of a pediatric hospital in Uganda, which demonstrated significant changes to both process and clinical outcomes. The initiative resulted in high triaging rates and a sustained reduction in time-to-antimicrobials”.

If the goal was “to sustain” triage rates, it should be reiterated that they were at 90% even before the implementation of the Smart Triage platform, and that triage and access rates had declined and then rose again during the COVID-19 period, as expected. The authors should discuss this issue.

• Lines 288-291: “Mortality of admitted patients decreased by almost half during this 5-year period, despite a sustained increase in patient numbers with comparable illness severity. Process improvements were the focus of this QI intervention; however, the ultimate purpose of all QI efforts was the improvement of clinical outcomes”.

Association between severity and mortality is ecologically based. It would be important to know the causes of death to assess the plausibility of the proposed association between QI and outcome; are they the same, have they changed? Are they causes that might actually be affected by triage methodology? The authors themselves raise the question: “While decreased mortality rates due to sepsis have been linked to a decreased time-to antimicrobials, this is likely also related to changes in clinical awareness and other elements of the sepsis bundle of care, such as intravenous fluid resuscitation”. The authors should discuss this issue.

• Lines 298-302: “This change is likely to be highly correlated to other simultaneous QI initiatives. For example, HICH prioritized the education of their nursing staff during this intervention timeframe and supported nurses in receiving diploma level training. HICH also mandated the consistent supply of medicines on their site, thereby eliminating treatment delays that were due to stockouts”.

Would it be possible to have some more information about the organization and staff structure, as done, for example, by Onyango et al. in the cited article (reference 11)?

• Lines 303-304: “A stable triage rate above 90 percent is indicative of staff adoption and adherence to the QI process”.

The triage rate started from 90%. Anyway, why would the triage rate decrease without QI?

• Lines 306-307: “The increased volume of patients is likely to be due to the efficient and effective service provided”.

Do the authors mean that the hospital has become more attractive? do they attribute this to the implementation of smart triage or to the overall improvement?

• Lines 308-310: “The QI interventions were sustained even with increasing OPD attendance. This may be attributable to an increase in funding at HICH to support additional staffing, expand infrastructure and improve workflow to accommodate the increased demands”.

As mentioned earlier, can you get some more data about the organization and staff structure changes? In fact the authors stated “It is unlikely that the decrease in time to antibiotics of 24 minutes is solely responsible for better outcomes; better triage and attention to overall care may be responsible but difficult to prove”.

• Lines 324-328: “Ideally, antimicrobial agents should be administered to those who are likely to benefit and withheld from those who do not need it. However, identification of groups likely to benefit is difficult in the absence of microbial and laboratory support. Lack of laboratory support also hampers antimicrobial stewardship programs. That there was a decrease in mortality rate despite less antimicrobial use is heartening, but difficult to explain”.

Has the use of antimicrobials decreased? How come there is no laboratory support? Why are we not shown data on this? Could the reduction in administration time depend on the decreased use of therapy, rather than the Smart Triage platform implementation?

Moreover, later in the text (lines 348-349) it says: “Lab personnel also began learning QI strategies and have initiated lab specific QI projects with the Smart Triage QI Team’s guidance”. Can the authors clarify whether or not there was laboratory support?

• Lines 334-340: “Many QI initiatives are criticized because QI initiatives are often completed in siloes and do not lead to sustainable organizational-level change. Continuous QI (CQI) embeds itself into institutional culture in healthcare by fostering a culture of continuous learning, innovation, and improvement”. This QI initiative demonstrates that HICH’s commitment to continuous QI led to sustainable change and impacted long-term health outcomes”.

The fact that these are isolated and unstructured initiatives may be an effective limitation. But above all, perhaps the use of the verb “to demonstrate” should be avoided, also in view of what is stated in the following paragraph “Challenges in implementation”.

Challenges in implementation

• Lines 367-372: “Healthcare workers and patients communicated concerns regarding the tracking of patient locations and concerns about the health effects of wireless energy waves. Multiple QI initiatives were aimed at reducing beacon loss, which occurred at the expense of other potential initiatives. Treatment tracking using the beacons was difficult to maintain and beacons were eventually phased out of HICH’s workflow with the implementation of a new electronic charting system”.

So the hospital is abandoning the system they presented in the study? Can this point be clarified?

• Lines 372-382: “While the automatic tracking of treatments is desirable and wireless technologies are widely used for tracking in many applications, the barriers to adoption in this context were significant and were not reliably overcome. Smart Triage can be implemented without the Smart Spot system. Smart Triage relies on the use of a phone or tablet with connected pulse oximeter for optimal functionality. Any loss or malfunction of the pulse oximeter impacts data quality and accuracy. Vital signs can be entered from other available devices; however, this may degrade reliability. Equipment maintenance and replacement due to theft and wear-and-tear must be considered to ensure sustainability. Disruptions to the local network also cause issues with uploading patient data and using the clinical dashboard, but the platform is not reliant on an outside Internet connection which can more frequently be unstable”.

All these considerations undermine previous claims about the long-term success and sustainability of the initiative. The authors should discuss this issue.

Conclusions

• Lines 390-394: “Throughout the course of this initiative, we demonstrated a sustained improvement in triaging rates and time-to-antimicrobials in HICH’s OPD through multiple simultaneous PDSA cycles that targeted OPD staff training and OPD workflows while fostering a QI culture within the OPD. The decrease in mortality rates for admitted patients at HICH indicates the real impact of this initiative”.

Such statements are too strong, because they are obtained through “ecological” associations without any possibility of analytically assessing (age, clinical conditions at access) other factors potentially associated with the outcomes.

6. PLOS authors have the option to publish the peer review history of their article (what does this mean? ). If published, this will include your full peer review and any attached files.

**Do you want your identity to be public for this peer review?** For information about this choice, including consent withdrawal, please see our Privacy Policy .

Reviewer #1: No

Reviewer #2: No

---

## [Author Response · Author response to Decision Letter 1]

26 May 2025

Re: Improving pediatric care in Uganda with a digital platform and quality improvement initiative: a retrospective review of Smart Triage + QI

We would like to thank the reviewers for their insightful and thorough review of the manuscript. We address each of their points below. Each point is presented on a light grey background; our response is underneath, with manuscript text in italics and any additions to the text in bold.

Many of these comments have prompted modifications to the manuscript text. A revised version has been submitted, with tracked/highlighted changes along with a final version, as requested.

We hope that we have addressed the reviewers’ questions adequately and that you find our proposed revisions to the manuscript acceptable.

Sincerely,

Dr J Mark Ansermino

Department of Anesthesiology, Pharmacology & Therapeutics | Institute for Global Health

The University of British Columbia | BC Children's Hospital

938 W 28th Ave | Vancouver BC | V5Z 4H4 Canada, Phone 604 997 7753, mansermino@bcchr.ca

Reviewer #1

Title: Improving pediatric care in Uganda with a digital platform and quality improvement initiative: a retrospective review of Smart Triage + QI.

General comment: The smart Triage and QI were implemented in the OPD. The results on mortality reduction observed in inpatients may not not necessarily be accounted for solely by interventions in the OPD.

We agree that the improved outcomes may not be due solely to interventions in the OPD. Our project quality improvement efforts included the OPD, as well as early resuscitation and other areas such as radiology, laboratory, and pharmacy. The interactions between multifaceted changes in care (often called bundles of care) intended to improve outcomes are complex and optimal outcomes are achieved with the adoption of the entire bundle. Assigning relative value to individual components is challenging in quality improvement programs because it is difficult to assess the contribution of each component of the bundle separately.

We have added this to the discussion [p18]

The interactions between multifaceted changes in care (often called bundles of care) in this intervention are complex, and optimal outcomes are achieved by adopting the entire bundle. Assigning relative value to individual components is challenging because it is difficult to assess the contribution of each element of the bundle separately.

Abstract:

Ordinarily discussion is not acccommodated in the abstract instead a conclusion is more appropriate.

We have changed the title of the section to “conclusion” and shortened the text to be more appropriate for a conclusion.

Introduction:

Smart Triage is mentioned briefly wheraas QI is ommitted in the introduction.

We have added a section to the introduction [p4] on the importance of QI:

The quality of healthcare is suboptimal in many low and middle-income countries. More than 8 million people per year die from conditions that should be treatable by the health system (16): nearly 60% of these deaths are preventable but occur due to poor quality of care. Indeed, the mortality burden attributable to suboptimal care is estimated to be larger than that due to lack of access to care. Hence, quality improvement (QI) efforts are vital, but they must take a holistic approach based on the local context and resource constraints in those settings.

Ref 16. Kruk et al. High-quality health systems in the Sustainable Development Goals era: time for a revolution. Lancet Glob Health. 2018;6(11):e1196–252.

Results:

Figures ought to have legends/titles on the bottom.

We have added the details for the legends.

Discussion:

Whereas there was an observed reduction of mortality during the review perod my concern is to link it directly to the interventions in OPD unless if the QI was hospitalwide. The Authors should first focus on traige rates and time to antimicrobials and can only state that the interventions could have contributed to the reductions in mortality seen.

We agree that the reduction in mortality observed cannot be attributed solely to the interventions in the OPD. We have recently published a large, multi-country, multicenter trial that showed a mortality reduction due to the implementation of the Smart Triage platform and hospital-wide QI initiatives:

Ansermino et al. Implementation of Smart Triage combined with a quality improvement program for children presenting to facilities in Kenya and Uganda: An interrupted time series analysis https://doi.org/10.1371/journal.pdig.0000466.

The goal of this paper is to evaluate the sustainability of the program over a protracted period beyond the conclusion of our research project. We know that many initiatives, especially those in QI, are difficult to sustain beyond the end of the research. We have added the following text to the opening paragraph of the Discussion [p16] to reflect the reviewer's concerns:

However, other factors, such as vaccination rates, case mix, and other facility-wide quality improvements, may have contributed to this mortality reduction.

We have also added a limitations section [p20] and adjusted the conclusion [p21] to reflect the effect of other confounders on the mortality reduction (new text in bold):

Limitations

The major limitation of this study is that multiple data sources were used, each with their own shortcomings. We chose to use the monthly hospital mortality reports as our study data was focused on triage and treatment tracking, rather than mortality, and access to the EMR was limited to patient counts. Not all patients were recruited to the study and the observation period covered more than one study. The mortality reduction observed was likely not due to the Smart Triage intervention alone. There may have been sustained improvements in early resuscitation and other areas such as radiology, laboratory, and pharmacy beyond the research period. In addition, it is possible that other interventions, such as improved vaccination rates, improved healthcare seeking or improvements in different areas of the facility, may account for the sustained mortality reduction.

Conclusion

We observed a sustained improvement in triaging rates and time-to-antimicrobials in HICH’s OPD through multiple simultaneous PDSA cycles that targeted OPD staff training and OPD workflows while fostering a QI culture within the OPD. The decrease in mortality rates for admitted patients at HICH suggests the real impact of this initiative. Still, we cannot rule out other changes in the delivery of health services as contributors. The success of digital health in low-resource settings relies on the consistent commitment of a leadership team and the continuous adaptation to its environment; this supports the continuous improvement of quality of care and the goal of universal health coverage.

Conclusion:

Authors are expected to write concise statements in the conclusion section. References are not expected in this section.

The conclusion has been shortened and references removed (see our response to your previous comment)

Reviewer #2

The study was focused on the feasibility and implementation of the Smart Triage, within the Quality Improvement initiative in the outpatient department (OPD) of a pediatric hospital in Uganda, over a 5-year period (2018-2023). Smart Triage joins a risk prediction algorithm and a digital platform to enable healthcare workers to triage patients and track treatments effectively.

The authors stated that “the aim of QI initiative was to improve triaging rates and to reduce time-to antimicrobials in the hospital’s outpatient department (OPD)”. Moreover, in the section “QI intervention” they declared: “QI efforts were aimed at increasing the triaging rate to 100% in the OPD and reducing the time-to-antimicrobials to below a median of 45 minutes to reduce the number of patients waiting for longer periods”. Finally, at the end of the “Introduction”, the authors affirm: “This paper reports the most significant system-level changes that contributed to sustained improvements in these metrics”.

The study is interesting, but there are some critical issues that would be important to clarify.

Thank you. We have responded to each of your comments below.

• Smart Triage is described in the Introduction (lines 69-77) and in the Smart Triage Platform section (lines 107-129) as a digital platform that consists of three components: 1) a smart triage mobile application; 2) a smart spot bluetooth low energy system; 3) a clinical dashboard. All the Smart Triage system is summarized in figure 1. However, can the authors better clarify how exactly beacons work?

The following text has been added [p6] to describe the beacon system in more detail (new text in bold):

The Smart Spot BLE system consists of color-coded lanyards, based on patients’ triage categories, with an attached BLE patient beacon, treatment beacons, and BLE readers. At the end of treatment, the beacon number (labelled on the beacon) is entered with the patient’s data and sent to the dashboard. The BLE readers were strategically located in the hospital to provide the patient’s location as they moved through the facility by detecting the signal strength from each beacon and communicating with the local server over Wi-Fi. Signal strengths exceeding a prespecified threshold were considered in that room (thresholds were identified based on room size and location in baseline testing). The three triage categories were non-urgent, priority, and emergency, and the respective patients are given green, yellow, and red lanyards to indicate their triage acuity score. Treatment beacons for each treatment type (IV antibiotics, IV fluids, oxygen, nebulization, and antimalarials) were located in the treatment room and are used to easily track the times-to-treatment as the difference between the arrival time and the time when the beacon button was pressed. The nurse administering the treatment pushes both the button on the patient's beacon and the button on the treatment beacon (within 10 seconds) to record a treatment. The Smart Spot system allows accurate and easy data collection for time-to-treatment, supporting a data-driven QI approach. Treatment times can also be manually added to the clinical dashboard.

• In the figure 2 “Key Driver Diagram for HICH” is shown the diagram summarizing the Smart Triage QI Team identification of three key drivers of triaging rates and wait time-to antimicrobials in the OPD. However, in the box “Aim” only the triaging rate objective is reported. Furthermore, the conceptual and graphical relationships between aim, key drivers, secondary drivers, and change ideas, should be better explained.

The project's overall aim was to reduce the time to antimicrobial administration. The key driver diagram was produced during the QI process. QI program priorities were decided by the local teams. This key driver diagram is only one example of what was developed during one PDSA cycle. During the study implementation, we identified that high triage rates were essential to ensure complete data collection. We noticed that the legends for the figures were not included. We have added the following text to the legend to address your concerns:

An example of a key driver diagram to optimize triage rates was developed during the quality improvement process. The diagram was used to target specific interventions, and their effectiveness was measured during the Plan-Do-Study-Act cycles. Similar diagrams were used for other quality improvement initiatives, such as waiting times in the laboratory or completion of vital sign measurements.

• In the figure 3 “HICH Smart Triage QI Interventions Timeline October 2018 - October 2023” is shown the time schedule of the activities implemented. It would appear that the implementation of the smart triage mobile application took place between March 2022 and October 2023. However, further on in the manuscript (lines 228-229) it says: “November 1, 2018, the start of the first full month with the triage platform”. Since the smart triage mobile application would seem indispensable to the triage platform, would it be possible to clarify this apparent time contradiction?

We apologize for the confusion! There were two separate studies. The first was the ‘PocketDoc’ study, a platform feasibility study that included an earlier version of the mobile application and a short period of QI training. ‘Smart Triage’ was a follow-on project with updates to the app to study adding formal QI Training, which was implemented as a QI project.

We have now included a new table that describes each data source used (Table 1). We have added clarification in the legend of Figure 3 and the figure itself:

The platform feasibility study was conducted between 2018 and 2021, and a formal quality improvement study was conducted from 2021 to 2023.

• A more relevant issue is about “Data sources”. The authors affirm (lines 214-218): “For this analysis, data was collected from HICH’s monthly reports (aggregate data), the Smart Triage platform (deidentified data) and the Electronic Medical Records (deidentified data). HICH is mandated to provide monthly reports to the Ugandan Ministry of Health, including reports of mortality rates and admission rates”.

Can the authors clarify what deidentified data are? If they were individual data, why were all analyses instead conducted on aggregate data derived from the hospital's monthly reports? Was it conceivable to construct an individual-based dataset from the Electronic Medical Records, which would have allowed analytically, not ecologically, to assess associations between certain risk factors and the outcomes studied?

Thank you for this question. Ideally, it would have been more efficient if we were able to conduct all analyses on aggregate data derived from the hospital's monthly reports. A significant limitation was that mortality rates were unavailable in the study data. In addition, the study's outcome data collection was restricted to patients included in the study. To standardize the outcome measure, we chose to use the hospital-reported outcome data, which was more comprehensive than the study records. We were not able to link the specific death to the patient in the study. We have now added a new table that clarifies the sources of all data collected (Table 1).

We have clarified this issue in the limitations of the study [p20]:

Limitations

The major limitation of this study is that multiple data sources were used, each with their own shortcomings. We chose to use the monthly hospital mortality reports as our study data was focused on triage and treatment tracking, rather than mortality, and access to the EMR was limited to patient counts. Not all patients were recruited to the study and the observation period covered more than one study. The mortality reduction observed was likely not due to the Smart Triage intervention alone. There may have been sustained improvements in early resuscitation and other areas such as radiology, laboratory, and pharmacy beyond the research period. In addition, it is possible that other interventions, such as improved vaccination rates, improved healthcare seeking or improvements in different areas of the facility, may account for the sustained mortality reduction.

Evaluation and analysis

• Lines 220-222: “HICH’s primary outcomes were the triaging rates within the OPD, the median time from triage to initiation of antimicrobials (minutes), and the mortality rates of admitted patients at HICH”.

Why was the evaluation of mortality rates not stated early on as one of the objectives of the study?

The individual studies' numbers were too small to expect a significant decrease in mortality, and the studies were not powered to demonstrate a reduction in mortality.

• Lines 224-226: “We began collecting baseline data with the initial launch of PocketDoc in August 2018; therefore, we have included this data when possible, to establish a strong baseline for our data analysis”.

What do the authors mean by “therefore, we have included this data when possible”? What data might not have been included?

We apologize for the confusion created by this statement. We have clarified in the text [p11, see below, new text i

---

## [Decision Letter · Decision Letter 1]

PONE-D-24-57615R1Improving pediatric care in Uganda with a digital platform and quality improvement initiative: a retrospective review of Smart Triage + QIPLOS ONE

Dear Dr. Ansermino,

Thank you for submitting your manuscript to PLOS ONE. After careful consideration, we feel that it has merit but does not fully meet PLOS ONE’s publication criteria as it currently stands. Therefore, we invite you to submit a revised version of the manuscript that addresses the points raised during the review process.

We look forward to receiving your revised manuscript.

Kind regards,

Tai-Heng Chen, M.D., Ph.D.

Academic Editor

PLOS ONE

Journal Requirements:

Reviewers' comments:

Reviewer's Responses to Questions

**Comments to the Author**

1. If the authors have adequately addressed your comments raised in a previous round of review and you feel that this manuscript is now acceptable for publication, you may indicate that here to bypass the “Comments to the Author” section, enter your conflict of interest statement in the “Confidential to Editor” section, and submit your "Accept" recommendation.

Reviewer #2: (No Response)

2. Is the manuscript technically sound, and do the data support the conclusions?

Reviewer #2: Yes

3. Has the statistical analysis been performed appropriately and rigorously? 

Reviewer #2: Yes

4. Have the authors made all data underlying the findings in their manuscript fully available?

Reviewer #2: Yes

5. Is the manuscript presented in an intelligible fashion and written in standard English?

Reviewer #2: Yes

6. Review Comments to the Author

Reviewer #2: The new version of the manuscript clarifies almost all the doubts we had raised.

However, there are a couple of observations that we suggest to the authors that could further improve their manuscript.

• The authors state that they cannot analyse specific mortality rates for patient characteristics, as they do not have a data source that allows linkage with the patients included in the study. However, in response to a question about a statement at the beginning of the paragraph Evaluatyon and analysis

'Lines 220-222: "HICH's primary outcomes were the triaging rates within the OPD, the median time from triage to initiation of antimicrobials (minutes), and the mortality rates of admitted patients at HICH”. Why was the evaluation of mortality rates not stated early on as one of the objectives of the study?

the authors answer: “The individual studies' numbers were too small to expect a significant decrease in mortality, and the studies were not powered to demonstrate a reduction in mortality".

It is therefore unclear whether mortality among the subjects studied was not calculated due to low numbers of observed events or whether it was not among the objectives of the study anyway, since the study was not sized to assess it. It would be interesting to understand this, since mortality rates of admitted patients at HICH is counted among the primary outcomes of the study.

• The authors responded to the observation that “The OPD triage rate at the end of the observation period is broadly similar to that in 2018, before the implementation of the Smart Triage platform” (about 90%) that “The reason for the high triage rate at the start of the observation is that the triage was supported by the study team during the baseline period”. They modified the text accordingly. Would it be possible to have an estimate of the triage rate before the start of this support before the baseline period?

7. PLOS authors have the option to publish the peer review history of their article (what does this mean? ). If published, this will include your full peer review and any attached files.

**Do you want your identity to be public for this peer review?** For information about this choice, including consent withdrawal, please see our Privacy Policy .

Reviewer #2: **Yes: ** Anteo Di Napoli

---

## [Author Response · Author response to Decision Letter 2]

25 Jun 2025

We want to thank the reviewer for their comments. We appreciate your efforts to improve the manuscript. We address each of their points below. Each point is presented on a light grey background; our response is underneath, with manuscript text in italics and any additions to the text in bold.

We have made two additional modifications to the manuscript.

We hope that we have addressed the reviewers’ questions adequately and that you find our proposed revisions to the manuscript acceptable.

Sincerely,

Dr J Mark Ansermino

Department of Anesthesiology, Pharmacology & Therapeutics | Institute for Global Health

The University of British Columbia | BC Children's Hospital

938 W 28th Ave | Vancouver BC | V5Z 4H4 Canada, Phone 604 997 7753, mansermino@bcchr.ca

Reviewer #2: The new version of the manuscript clarifies almost all the doubts we had raised.

However, there are a couple of observations that we suggest to the authors that could further improve their manuscript.

• The authors state that they cannot analyse specific mortality rates for patient characteristics, as they do not have a data source that allows linkage with the patients included in the study. However, in response to a question about a statement at the beginning of the paragraph Evaluatyon and analysis

'Lines 220-222: "HICH's primary outcomes were the triaging rates within the OPD, the median time from triage to initiation of antimicrobials (minutes), and the mortality rates of admitted patients at HICH”. Why was the evaluation of mortality rates not stated early on as one of the objectives of the study?

the authors answer: “The individual studies' numbers were too small to expect a significant decrease in mortality, and the studies were not powered to demonstrate a reduction in mortality".

It is therefore unclear whether mortality among the subjects studied was not calculated due to low numbers of observed events or whether it was not among the objectives of the study anyway, since the study was not sized to assess it. It would be interesting to understand this, since mortality rates of admitted patients at HICH is counted among the primary outcomes of the study.

____

Thank you for your insightful review of the results! As previously indicated, two prospective studies were conducted, followed by a subsequent retrospective quality improvement project, which is reported in this manuscript. We do have mortality data in the two prospective studies; however, this is limited to the subjects recruited during the study periods. The study team does not have data on the patients who were not recruited, those seen between the two studies and after the second study. The mortality data we report in this manuscript is from the facility mortality reports. This is limited to in-hospital mortality data.

We have clarified this in the manuscript.

Line 239-242: HICH’s primary outcomes were the triaging rates within the OPD, the median time from triage to initiation of antimicrobials (minutes), and the mortality rates of admitted patients from the monthly facility mortality reports at HICH (mortality data for triaged patients was only determined during the study periods).

Line 266-269: The overall trajectories of change of OPD attendance, triaging rates, illness acuity (independent ED-PEWS urgency categories), time-to-antimicrobials, and mortality rates from the facility monthly statistics of all patients were assessed using simple linear regression with time as the explanatory variable.

The authors responded to the observation that “The OPD triage rate at the end of the observation period is broadly similar to that in 2018, before the implementation of the Smart Triage platform” (about 90%) that “The reason for the high triage rate at the start of the observation is that the triage was supported by the study team during the baseline period”. They modified the text accordingly. Would it be possible to have an estimate of the triage rate before the start of this support before the baseline period?.

In this facility, as in many facilities in LMICs, there was no formal triage process in place before the feasibility study in 2018. While the WHO ETAT guidelines were used, this does not dictate a formal process of seeing patients in a specific order. The nurse who would perform the initial observation may determine that the patient was critically ill, but this information was not recorded.

---

## [Decision Letter · Decision Letter 2]

Improving pediatric care in Uganda with a digital platform and quality improvement initiative: a retrospective review of Smart Triage + QI

PONE-D-24-57615R2

Dear Dr. Ansermino,

We’re pleased to inform you that your manuscript has been judged scientifically suitable for publication and will be formally accepted for publication once it meets all outstanding technical requirements.

Kind regards,

Tai-Heng Chen, M.D., Ph.D.

Academic Editor

PLOS ONE

Additional Editor Comments (optional):

Reviewers' comments:

Reviewer's Responses to Questions

**Comments to the Author**

1. If the authors have adequately addressed your comments raised in a previous round of review and you feel that this manuscript is now acceptable for publication, you may indicate that here to bypass the “Comments to the Author” section, enter your conflict of interest statement in the “Confidential to Editor” section, and submit your "Accept" recommendation.

Reviewer #2: All comments have been addressed

2. Is the manuscript technically sound, and do the data support the conclusions?

Reviewer #2: Yes

3. Has the statistical analysis been performed appropriately and rigorously? 

Reviewer #2: Yes

4. Have the authors made all data underlying the findings in their manuscript fully available?

Reviewer #2: (No Response)

5. Is the manuscript presented in an intelligible fashion and written in standard English?

Reviewer #2: Yes

6. Review Comments to the Author

Reviewer #2: The authors also clarified the last doubts. I believe the manuscript will be very useful for researchers and public health practitioners.

7. PLOS authors have the option to publish the peer review history of their article (what does this mean? ). If published, this will include your full peer review and any attached files.

**Do you want your identity to be public for this peer review?** For information about this choice, including consent withdrawal, please see our Privacy Policy .

Reviewer #2: **Yes: ** Dr Anteo Di Napoli; MD, MSc
